# Effectiveness of Physiotherapy in Patients with Ankylosing Spondylitis: A Systematic Review and Meta-Analysis

**DOI:** 10.3390/healthcare10010132

**Published:** 2022-01-10

**Authors:** Luca Pontone Gravaldi, Francesca Bonetti, Simona Lezzerini, Fernando De Maio

**Affiliations:** Department of Clinical Sciences and Translation Medicine, University of Rome “Tor Vergata”, 00133 Rome, Italy; luca.pontonegravaldi@libero.it (L.P.G.); simona.lezzerini@uniroma2.it (S.L.); demaio@med.uniroma2.it (F.D.M.)

**Keywords:** physical therapists, spondylitis, ankylosing, physical therapy modalities, exercise therapy, exercise

## Abstract

This study aimed to evaluate the safety and effectiveness of non-pharmacological interventions supervised by a physiotherapist in patients with Ankylosing Spondylitis, PROSPERO Protocol number CRD42020209453. Five databases (PubMed, PEDro, Scopus, Web of Science Core, and EMBASE) and reference lists with relevant articles were searched. Randomised controlled trials (RCTs) on the effectiveness of non-pharmacological interventions supervised by a physiotherapist were compared with usual care or home-based exercise programmes. Two investigators independently screened eligible studies. A total of 12 RCTs satisfied eligible criteria. The risk of bias ranged between medium and high. The meta-analysis results indicated that between supervised physiotherapy and usual care, the former was significantly associated with improvement in disease activity (standardised mean difference = −0.37, 95% CI, −0.64; −0.11; *p* < 0.001, I^2^ = 71.25%, *n* = 629), and functional capacity (standardised mean difference = −0.36, 95% CI, −0.61; −0.12, *p* < 0.05; *n* = 629). No statistically significant differences emerged when interventions were compared with home-based exercise programmes. Supervised physiotherapy is more effective than usual care in improving disease activity, functional capacity, and pain in patients with ankylosing spondylitis. No significant improvements emerged when supervised physiotherapy and home-based exercise programmes were compared. Further investigation and RCTs with larger samples are needed.

## 1. Introduction

Ankylosing spondylitis (AS) is a painful and progressive chronic inflammation of the axial skeleton that mainly affects the spine and sacroiliac joints [1]. Over time, because of the fusion of some small bones, the spine can become less flexible and result in a hunched-forward posture [2]. The AS prevalence rate is estimated between 0.03 and 1.8% in Europe, North America, and China [3]. The incidence ranges from 0.5 to 14 per 100,000 people per year, depending on the country [4]. The male–female ratio, around 3:1, shows that it is more common among men than women [5].

Many comorbidities, chronic pain, functional disabilities, and resource consumption are associated with ankylosing spondylitis. Thus, the management of AS may result in high direct and indirect costs to the patients and the healthcare system [6,7]. AS has a mostly idiopathic aetiology. To date, the human leukocyte antigen HLA-B27 has been suggested as a key element in the pathogenesis of AS [8]. AS prevalence in people with positive (HLA)-B27 is approximately 5–6% [9]. Genetic studies have also demonstrated that HLA-B27 contributes to ~20.1% of AS heritability [10]. First symptoms usually occur before 30 years and seldom after 45 years [11]. The modified New York (mNY) criteria, Amor criteria, European Spondyloarthropathy Study Group (ESSG), and ASAS (Assessment of SpondyloArthritis International Society, 2016) criteria are frequently used to assess AS diagnosis [12,13,14]. The main clinical manifestations of AS are back pain and progressive spinal rigidity, as well as inflammation of the hips, shoulders, peripheral joints, and fingers/toes. Furthermore, inflammatory skin conditions, inflammatory bowel disease, enthesitis, and anterior uveitis can also be present [15].

The management of AS aims to improve and maintain spinal flexibility and normal posture, relieve symptoms, decrease functional limitations, and reduce complications [16]. In 2011, ASAS and EULAR (European League Against Rheumatism) updated the existing EULAR management recommendations for AS. A combination of non-pharmacological and pharmacological treatments has been recommended to reduce patients’ discomfort [17]. The mainstays of pharmacological treatment involve non-steroidal anti-inflammatory medications (NSAIDs) and TNF-α inhibitors (TNFis). Other therapies include non-TNFi biologics (secukinumab), methotrexate, and sulfasalazine. Approximately 20–40% of patients do not respond well to pharmacological treatment [18,19]. Exercising regularly and educating patients are the cornerstones of non-pharmacological treatment to reduce symptomatology [15]. The effectiveness of exercise might depend on the individual’s adherence to the prescribed programme, considering the dose–response relationship between exercise and health effects [20,21,22]. Therefore, physiotherapy plays a crucial role in the management of AS. Patient training and exercise programmes supervised by a physiotherapist may improve symptoms and teach patients how to independently and adequately handle AS throughout life, thereby reducing the cost impact of physiotherapy [23]. To date, a specific non-pharmacological protocol is not yet available [24], and the effect of different types of exercise programmes remains unclear [25].

Many systematic reviews with meta-analysis were carried out about the role of exercise programs in improving AS symptomatology [24,26,27] including RCTs or uncontrolled trials. However, none of them analysed the role of supervised physiotherapy, which plays a crucial role in the non-pharmacological management of AS. Supervised physiotherapy is a process where knowledge is transferred through instruction, demonstration, and reflection [28]. In this process, patients learn what to do and how to implement exercise while sharing feelings and issues with their physiotherapist [29]. This creates a special physiotherapist–patients trust relationship that could modify the patients’ perception toward the effectiveness of exercise programs [30]. 

This systematic review with meta-analysis aims to understand and contribute to the evidence about the safety and effectiveness of non-pharmacological interventions in patients with AS. More specifically, its main goal is to assess the effectiveness of non-pharmacological supervised interventions in improving physical function, disease activity, spinal mobility, reducing pain, and enhancing the quality of life in patients with AS. The review question was: Is supervised physiotherapy effective in reducing symptoms and improving quality of life compared to home-based exercise programs or usual care in patients with AS? We hypothesised that supervised physiotherapy reduces disease activity and pain while improving functional capacity, spinal mobility, and quality of life. 

The components of the PICO question were: (Patients) patients with AS; (Intervention) non-conventional exercises supervised by physiotherapists, balneotherapy, or spa therapy; (Comparison) usual care, conventional exercises, or home-based exercise programmes; (Outcome) Bath Ankylosing Spondylitis Metrology Index (BASMI), Bath Ankylosing Spondylitis Functional Index (BASFI), Bath Ankylosing Spondylitis Disease Activity Index (BASDAI), Visual Analog Scale score for pain (VAS) and AS-specific measure of Quality of Life (ASQoL).

## 2. Materials and Methods

The material and methods are based on the PRISMA (Preferred Reporting Items for Systematic Reviews and Meta-Analyses) guidelines. The methodology was previously registered in the PROSPERO (International prospective register of systematic reviews) database under the protocol number CRD42020209453.

### 2.1. Data Source and Search Strategy

A comprehensive search strategy was used on PubMed, PEDro, Scopus, Web of Science, and EMBASE to identify all relevant studies irrespective of language or publication status. The search strategy was performed until 27 November 2020 and updated in December 2021. Duplicates were manually removed. After selecting articles, we checked the reference lists of all identified RCTs, other relevant papers, and significant English and Italian textbooks about AS. The search strategy was reported in Appendix A.

### 2.2. Inclusion Criteria and Study Selection

Studies were included if they strictly met the following criteria: (1) patients with a diagnosis of AS according to modified New York Criteria, Amor Criteria or ASAS criteria for axial spondylarthritis; (2) randomised controlled trials (RCTs) and published and unpublished trials; cross-over trials; cluster RCTs with assessments of their potential for unit of analysis error; (3) non-pharmacological interventions as physiotherapy, exercise programmes, balneotherapy, spa therapy, education, or self-education group; (4) extractable data on BASMI, BASFI, BASDAI (primary outcomes) and VAS or ASQoL (secondary outcomes). 

We removed duplicate records from the references identified. Two authors (LPG and FB) independently screened the search results using an over-inclusive approach. Titles and abstracts of citations identified from the search strategy were considered to construct a list of all potentially relevant papers. Full texts obtained for all articles had abstracts with a relevant title. Full inclusion criteria were applied to generate a list of studies. Two authors (LPG and FB) independently reviewed the list of studies, and those with a PEDro scale higher than six were included.

We had planned to resolve disagreements by consensus or discussions with a third review team member (AD) and reported this in the final review. However, no disagreement was present, and consequently, the kappa statistic was not reported. The selection process was documented by completing a PRISMA flow chart. 

### 2.3. Data Extraction and Management

Two review authors (LPG and FB) independently extracted data concerning first authors, publication years, study location, participant characteristics (sample size after interventions, mean age/age range), intervention details (physiotherapy intervention, home-based exercise programme, usual care/conventional treatment, co-intervention or alternative intervention), and outcome measures (BASFI, BASDAI, BASMI, VAS, ASQoL) for trials that met the inclusion criteria of the review. For each outcome, mean and standard deviation along with the number of participants were extracted for intervention and control groups. Where unavailable, we requested the original data from the authors of the RCTs. Those studies where the authors did not respond to us within three weeks were not included.

### 2.4. Assessment of Risk of Bias in Included Studies

We assessed the risk of bias using the tool described in the Cochrane Handbook for Systematic Reviews of Interventions. Seven specific domains were examined: sequence generation, allocation concealment, blinding of participants, blinding of personnel, blinding of outcome assessors, incomplete outcome data, and selective outcome, plus other potential sources of bias. Version 2 of the Cochrane risk-of-bias tool for randomised trials was used. If all domains were assessed at low risk for potential bias, studies were classified at low risk of bias; if one or more categories were assessed at high risk of bias, then the studies were at high risk of bias. If one or more key domains were assessed at unclear risk of bias or domain with some concerns, studies were classified as unclear/some concerns.

### 2.5. Data Synthesis

The STATA software (version 16; StataCorp LP, College Station, TX, USA) was used to synthesise study results on outcomes of interests. We analysed data separately by common group intervention: physiotherapy intervention versus usual care or conventional exercises (group I) and physiotherapy intervention versus home-based exercise programmes (group II). Synthesis of each included study was reported in Table 1. 

We calculated the mean difference (MD) between the groups since outcomes were continuously distributed. Where measures were reported using different scales, we calculated the standardised mean difference (SMD) if clinically appropriate. We also reported 95% confidence intervals (CI). The SMD is also known as Cohen’s d and is defined as the mean difference divided by the study’s standard deviation. Given a large clinical heterogeneity between physiotherapy interventions, participants, and characteristics, we used a random-effects model as it assumes the studies estimate different but related effects. Weighted averages of treatment effects were calculated by pooling SMD estimates across the studies using a random-effects model from the method of DerSimonian and Laird. Hedges’s g was used to calculate the effect size. 

We assessed heterogeneity using the I^2^ statistic and forest plots. According to the Cochrane Handbook for Systematic Reviews of Interventions, the bands of interpretation for I^2^ are as follows: 0–40% may be unimportant; 30–60% may represent moderate heterogeneity; 50–90% may represent substantial heterogeneity, and 75–100% may have considerable heterogeneity. We took values above 30% to indicate moderate heterogeneity and sought sources of heterogeneity. If moderate or more considerable heterogeneity emerged, we sought possible causes, including the demographic profile of the participants, the duration of treatment, or the exercise programmes’ combination. We used *p* values of 0.10 to indicate significant heterogeneity in the meta-analysis of small studies. A *p*-value lower than 0.025 was set for effective size significance evaluation after the Bonferroni adjustment.

### 2.6. Summary of Findings

Summary of findings tables were included to provide essential information about the quality of evidence, and the magnitude of the effect of the intervention in BASMI, BASDAI, BASFI, VAS, and ASQoL [31]. We assessed the overall quality of the evidence for each primary outcome by using the GRADE approach [32]. Summary of findings tables were developed through GRADEpro (GRADEpro GDT. Version accessed. Hamilton, ON, USA: McMaster University, developed by Evidence Prime, 2015. https://www.gradepro.org (accessed on 25 June 2021).

## 3. Results

### 3.1. Study Characteristics

Characteristics of the study are reported in Table 1. Twenty studies were considered [33,34,35,36,37,38,39,40,41,42,43,44]. Four were excluded because supervised physiotherapy was performed in the intervention and control groups (Figure 1).

Twelve RCTs were included in this study [33,34,35,36,37,38,39,40,41,42,43,44]. The earliest study was published in 2003, and the latest in 2019. Five studies were conducted in Turkey, two in Norway, two in Brazil, and one each in Switzerland, China, and Spain, respectively. In total, 1483 patients with AS (intervention group = 749; control group = 734) were found in the 12 selected RCTs. The mean age of participants ranged from 36 to 49.2 years old. The total number of training sessions ranged from 10 to 40, with a training duration from 2 to 12 weeks. The main activities were Pilates, intensive exercise programmes, Baduanjin Oigong exercise, and cardiovascular training. The group-based activity was the most frequent training mode. Eight studies had compared supervised physiotherapy with usual care, while four had compared supervised physiotherapy with home-based exercise programmes. 

### 3.2. Risk of Bias

The risk of bias assessment was reported in Figure 2. Fifty per cent and 41.7% of studies reported a high and moderate risk of bias, respectively. Only one study reported a low risk of bias. Deviation from intended interventions and selection of the reported results had the most frequent source of bias of included RCTs. All studies reported low risk for the randomisation process and missing outcome data. Three studies reported an intention to treat analysis [36,42,43].

According to “13ﬀ.3.5.4 tests for funnel plot asymmetry” [45], that recommends using tests for a funnel plot only when there are at least 10 studies included in the meta-analysis, not to reduce the power of the test, publication bias was not assessed.

### 3.3. Meta-Analysis of Outcome

A total of 12 RCTs examined the effectiveness of supervised physiotherapy in improving AS symptoms as measured by BASDAI, BASMI, BASFI, VAS, and ASQoL.

#### 3.3.1. BASDAI

Eleven studies examined the effect of supervised physiotherapy on disease activity (group I = 8, group II = 3) with lower scores indicating a reduction in disease activity. In group I, a significant benefit emerged in the intervention group after supervised training (SMD = −0.37, 95% CI, −0.64; −0.11; *p* < 0.001, I^2^ = 71.25%, *n* = 1246). No significant benefits in favour of the intervention group were reported in group II (SMD = −0.14; 95% CI, −0.42; 0.15; *p* > 0.05; I^2^ = 0.00, *n* = 191). No group differences emerged between the two groups [Q_b_ (1) = 1.48, *p* = 0.22]. Forest plots are reported in Figure 3. 

#### 3.3.2. BASMI

Eight studies examined the effect of supervised physiotherapy on spinal mobility (group I = 5; group II = 3). Lower scores indicate an increase in spinal mobility. In group II, a significant benefit emerged in the intervention group after supervised training (SMD = −0.20, 95% CI, −0.77; 0.37; *p* < 0.05, I^2^ = 74.58%, *n* = 191). (Figure 4). No group differences were noted [Q_b_ (1) = 0.06, *p* = 0.80]. 

#### 3.3.3. BASFIs

Twelve studies examined the effect of supervised physiotherapy on the degree of functional limitation (group I = 8; group II = 4). Lower scores indicate an increase in functional capacity. On comparing intervention and usual care, supervised physiotherapy significantly reduced functional limitation (SMD = −0.36, 95% CI, −0.61; −0.12; *p* < 0.05; *n* = 1247) (Figure 5). In the comparison between intervention and home-based exercise programmes, no statistical significance was noted (SMD = −0.29; 95% CI, −0.70; 0.12; *p* > 0.05; *n* = 236).

#### 3.3.4. VAS

Five studies examined the effect of supervised physiotherapy in reducing pain (group I = 3; group II = 2) (Figure 6). Lower scores indicate a reduction of pain. No significant asymmetries emerged for both groups (group I: β1 = 3.31, *p* = 0.7517; group II: β1= −3.00, *p* = 1.000). Supervised physiotherapy reduced pain compared to usual care (SMD = −0.31, 95% CI, −0.88; 0.25; *p* < 0.05; *n* = 813). No statistical significance was noted in the comparison between supervised physiotherapy and home-based exercise programmes (SMD = −0.27; 95% CI, −0.61; −0.07; *p* > 0.05; *n* = 166). 

#### 3.3.5. ASQoL

Four studies examined the effect of supervised physiotherapy in improving quality of life (group I = 3; group II = 1). Higher scores indicate a worse quality of life. In group II, only one study was included. In group I, supervised physiotherapy slightly improved ASQoL (SMD = −0.09; 95% CI, −0.51; 0.32; *p* = 0.04, *n* = 866) (Figure 7).

## 4. Discussion

Ankylosing Spondylitis (AS) is a common rheumatic disease that mainly affects young adults and produces many physical and psychological impediments which negatively impact everyday life. Many pharmacological and non-pharmacological treatments are available to reduce pain and stiffness in the back and sacroiliac joints and improve spinal and peripheral joint mobility [46]. However, the long-term management of AS necessitates a combination of drug treatment, physical therapies, and psychosocial interventions. Therefore, minimizing the impact of AS on both patients and the healthcare system is the main challenge. Supervised physiotherapy represents a crucial element in the management of AS because it may improve the effectiveness of exercise programmes and lead to a quick and more lasting reduction of AS symptomatology. This paper aimed to evaluate the beneficial effects of non-pharmacological interventions supervised by a physiotherapist in reducing AS symptoms.

Overall, the results indicated that supervised physiotherapy reduces disease activity and functional limitations and improves spinal mobility, while the effect of supervised physiotherapy on quality of life is unclear. Our work demonstrated an overall positive impact of supervised physiotherapy on disease activity and functional limitation compared to usual care. No effect emerged when comparing supervised interventions and home-based exercise programmes for disease activity and functional limitation, while a positive impact was registered for spinal mobility. In comparing supervised physiotherapy and usual care, five studies reported significant positive results in reducing disease activity. Pilates, exercise with videogames, cardiorespiratory and muscle strengthening exercises, rehabilitation programmes, educational training, and Stanger bath therapy positively affected pain, fatigue, swelling, and morning stiffness. The number of sessions, frequency, and mode of training did not appear to determine the efficacy of supervised therapy. Meta-analysis suggested a positive impact of supervised physiotherapy in improving spinal mobility compared to home-based exercise programmes. However, only one study out of four demonstrated significant positive results. Ultrasound treatment combined with supervised physiotherapy increased the effect of exercises, reducing patients’ physical limitations due to AS. 

We registered that supervised physiotherapy enhances the efficacy of regular physical exercise, giving patients a schedule to continue training. From our findings, it emerged that supervised physiotherapy allows patients to maintain an optimal mode of delivery, frequency and duration of the treatment, which, on the contrary, are not maintained with home-based exercise programs or usual care [47]: all indexes linked with the physical activity showed improvement, while other indexes, such as pain and quality of life, less linked with a physiotherapist’s supervision, did not report clear progress. Furthermore, continuous supervised training allowed patients to reduce AS symptoms such as disease activity and functional capacity and improve spinal mobility, positively affecting AS management. The positive impact of supervised physiotherapy was marked in the comparison between intervention vs. usual care than in the comparison between intervention vs. a home-based exercise program, where a type of remote supervision was already present. 

Moreover, our results confirmed what was already observed in previous studies: the efficacy of non-pharmacological interventions such as exercises, education, and physiotherapy in reducing AS symptomatology [26]. Regular exercise improves several outcomes, even though effects in disease activity are minor, such as functional and spinal mobility, especially in comparison with no intervention programme. A multimodal approach supervised by a physiotherapist and followed by a home-based regime was already suggested as an optimal disease management measure [48]. Patients’ frequency should also be a key component in AS management. Despite all patients being aware about the necessity of daily physical exercise, most of them tend to not participate in the exercise on a frequent basis [47]. The lack of information about this topic in the included studies prevented us from demonstrating a dose–response relationship. Quality of life did not report any improvement after supervised physiotherapy in accordance with several studies [49,50]. Mood changes, depression, and physical difficulties negatively impact the quality of life, with a higher rate in women than in men [51]. We concluded that supervised physiotherapy could reduce AS symptoms compared to usual care because experienced supervision allows patients to perform exercise programmes and adhere to long-term regimes correctly. An optimal disease management programme maintains spinal mobility and physical functions while controlling pain and inflammation [26,52]. The lack of significance in comparing intervention and home-based exercise programmes confirms the efficacy of any type of non-pharmacological treatment. Quality of life could be enhanced through group exercises and support groups, thus building positive experiences among patients [53].

Our review itself has some limitations. In some cases, we could not determine whether participants who had received usual care also had had exercises because some of the included studies poorly described the content of usual care interventions. Thus, participants in the usual care group could have practised exercises. In addition, a high heterogeneity among studies was detected. We tried to reduce statistical heterogeneity by grouping studies according to a control group (usual care or home-based exercise programmes), while for clinical heterogeneity, due to different types of intervention, number of sessions, and training mode (individual or group), we applied a random-effect model to address heterogeneity that cannot readily be explained given the low number of included studies.

Moreover, the lack of a sufficient number of studies in each group to perform a publication bias prevented us from determining the impact on our findings. Finally, we could have missed studies that are still unpublished or are currently being conducted. All included studies were RCTs. In some cases, assessing the quality of the trial was difficult because of a lack of information. Further, some of the included older studies did not reflect current methodological practices.

Many included studies reported significantly increased scores in pre-post analysis, especially in the intervention group, but failed to demonstrate a significant difference between groups. Except for one study [40], all RCTs were conducted on small sample size. The RCTs did not have a control group without any conventional training programme. However, this last issue could not be considered a limitation; since AS patients need to engage in physical exercises for their whole life, measuring experimentally the effectiveness of the total absence of physical activities could be inappropriate. The control groups were heterogeneous, thus affecting the power of the meta-analysis. Moreover, the prevalence of small studies could have lowered the change of detecting significance. Finally, since 44% of studies were conducted in one specific country, the generalisability of these study findings is limited by the ethnicity of study participants. 

We used the GRADE approach to assess the quality of the evidence examined for each outcome (Appendix A). Most of the evidence was downgraded to low or very low quality because of the risk of bias, inconsistency due to heterogeneity, and imprecision in small trials. Moreover, we did not perform subgroup analysis because of the small number of studies in each group. The presence of only one study at low risk of bias prevented us from executing a sensitivity analysis.

## 5. Conclusions

Our results confirm that a multimodal supervised approach through exercises, physiotherapy, education, balneotherapy, and other non-pharmacological interventions helps manage patients with AS. This work shows supervised physiotherapy is more effective than usual care in improving disease activity, functional capacity, and pain in patients with ankylosing spondylitis. 

Further studies are needed to investigate the main effects of exercises in the medium–long term and evaluate the effectiveness of mobilisation on the spine. These studies need to be carried out on larger samples. More incisive conclusions in the literature could improve AS patients’ everyday lives and reduce the health system’s expenditure.

## Figures and Tables

**Figure 1 healthcare-10-00132-f001:**
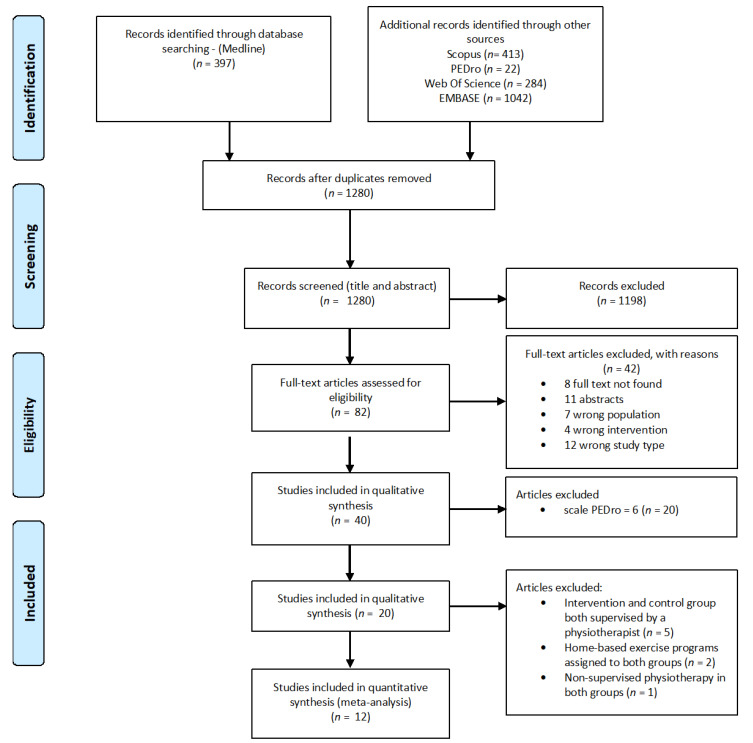
Prisma Flowchart.

**Figure 2 healthcare-10-00132-f002:**
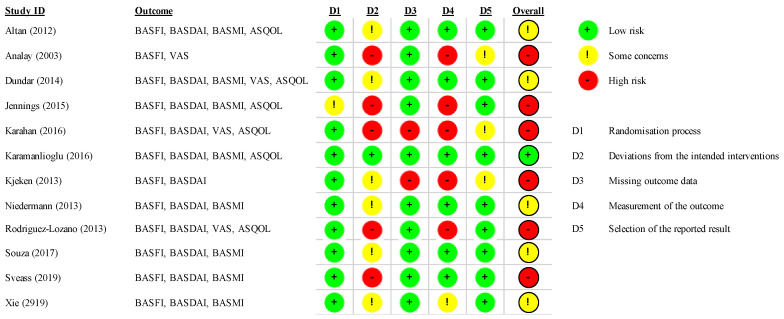
Risk of bias.

**Figure 3 healthcare-10-00132-f003:**
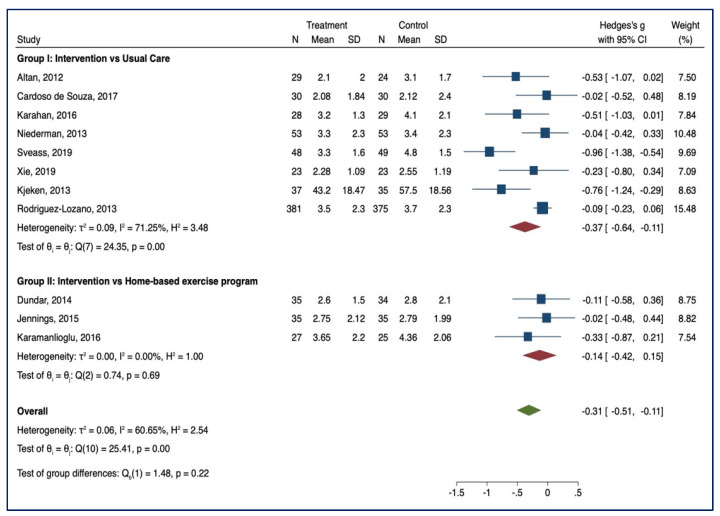
BASDAI—Forest Plot.

**Figure 4 healthcare-10-00132-f004:**
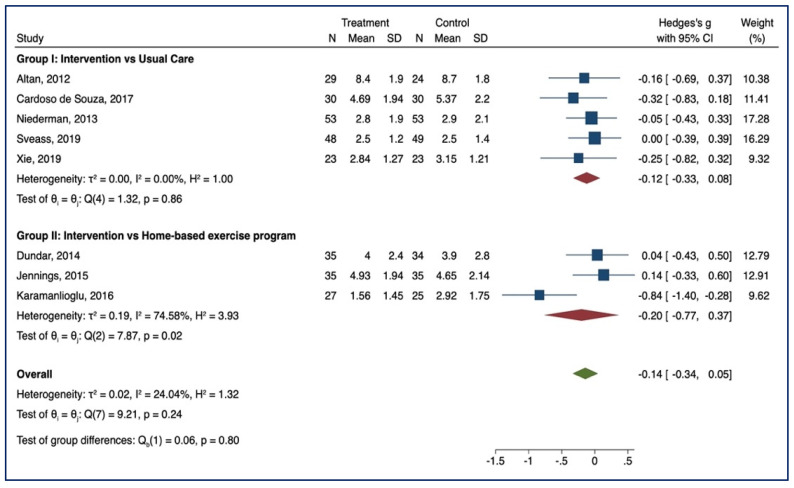
BASMI—Forest Plot.

**Figure 5 healthcare-10-00132-f005:**
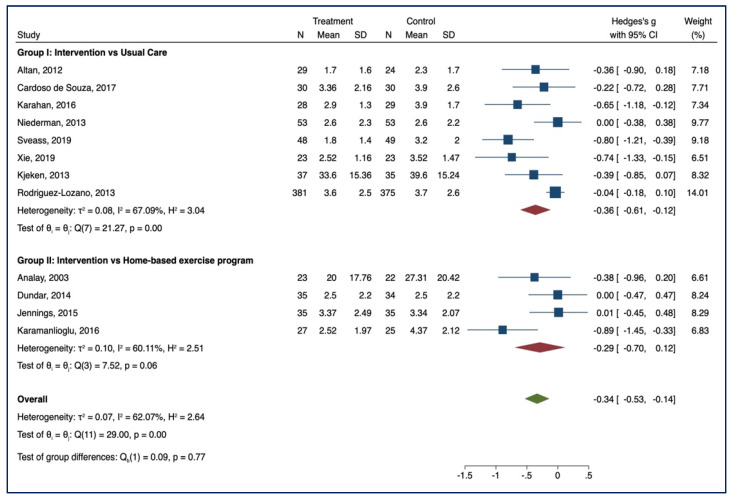
BASFI—Forest Plot.

**Figure 6 healthcare-10-00132-f006:**
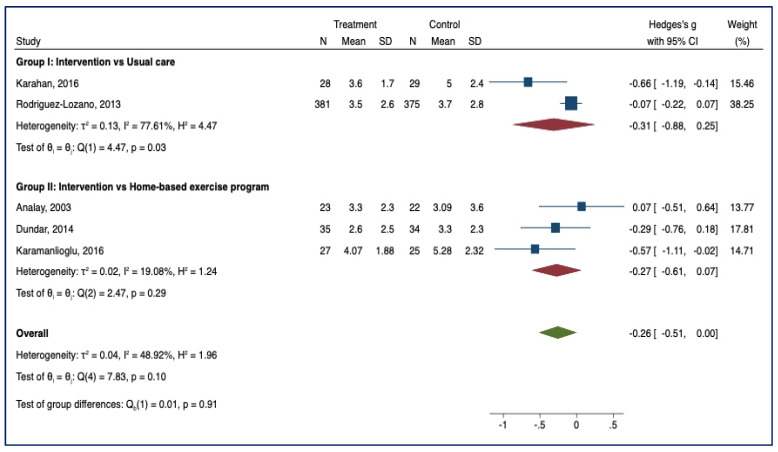
VAS—Forest Plot.

**Figure 7 healthcare-10-00132-f007:**
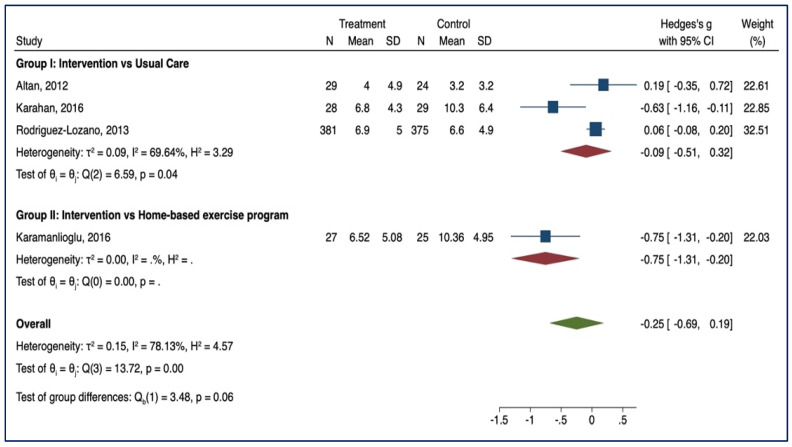
ASQoL—Forest Plot.

**Table 1 healthcare-10-00132-t001:** Study Characteristics.

First Author	Country	Year	Sample IG	Sample CG	Mean Age *	Exercise Description	Training Frequency	Mode of Combination	Training Mode	No. of Sessions	CG Activity	BASFI	BASDAI	BASMI	VAS	ASQoL	Main Results
**Altan**	Turkey	2012	29	24	45.23	Pilates	3 times a week for 12 weeks	Alternative	group	36	Usual care	yes	yes	yes	no	yes	Significant difference for BASDAI at 12 weeks (*p* < 0.01) and BASMI and BASFI at 24 weeks (*p* < 0.05) in the intervention group
**Analay**	Turkey	2003	23	22	36	Intensive exercise programme	3 times a week for 6 weeks	Simultaneous	group	18	Home-based exercise	yes	no	no	yes	no	Statistically significant difference in all parameters except pain (*p* < 0.05)
**Dundar**	Turkey	2014	35	34	42.7	Aquatic exercise	5 times a week for 4 weeks	Alternative	group	20	Home-based exercise	yes	yes	yes	yes	yes	Significant improvements for all parameters in both groups after treatment at week 4 and week 12 (*p* < 0.05)
**Jennings**	Brazil	2015	35	35	41.6	Aerobic and stretching exercise	3 times a week for 12 weeks	Alternative	individual	36	Stretching exercise	yes	yes	yes	no	yes	Significant improvement pre-post (*p* < 0.05), but no difference between groups
**Karahan**	Turkey	2016	28	29	36.4	Exercise with videogame	5 days a week for 8 weeks	Alternative	individual	40	Usual care	yes	yes	no	yes	yes	Significant differences between the two groups in VAS, BASFI, BASDAI and ASQoL; considerable improvement in the intervention group (*p* < 0.05)
**Karamanlioglu**	Turkey	2016	27	25	39.65	Ultrasound therapy and education programme	US: 10 sessions—Exercise: 5 times a week for 2 weeks	Alternative	individual	10	Instruction on exercise therapy	yes	yes	yes	no	yes	Significant results in intervention group for BASMI (*p* < 0.05) after 2 weeks and daily pain (*p* < 0.01), BASDAI (*p* < 0.05), and ASQoL (*p* < 0.05) after 6 weeks.
**Kjeken**	Norway	2013	37	35	49.2	Rehabilitation program	Pool: 3–5 sessions/week—Gym: 2–3 sessions/week—Outdoors: 3 sessions/week	Alternative	individual	NA	Usual care	yes	yes	no	no	no	Significant improvement in the intervention group for BASDAI (*p* < 0.05)
**Niederman ****	Switzerland	2013	53	53	48.9	Cardiovascular training	2 times a week for 12 weeks	Alternative	individual	24	Usual care	yes	yes	yes	no	no	After 3 months, significant improvement in the intervention group (*p* < 0.001)
**Rodriguez-Lozano**	Spain	2013	381	375	45.5	Education programme + exercise	2 h informative session	Alternative	group	NA	Usual care	yes	yes	no	yes	yes	After 6 months, significant difference in intervention group for BASDAI (*p* < 0.01), BASFI (*p* < 0.01), VAS (*p* = 0.02), and ASQoL (*p* < 0.01)
**Souza**	Brazil	2017	30	30	44.4	Exercise with Swiss ball	2 times a week for 16 weeks	Alternative	group	32	Usual care	yes	yes	yes	no	no	No significant differences between groups for BASFI and BASMI.
**Sveass**	Norway	2019	48	49	45.7	Cardiorespiratory and muscular strength exercise	2 times per week for 12 weeks	Alternative	individual	24	Usual care	yes	yes	yes	no	no	Significant improvement in the intervention group for BASDAI (*p* < 0.001) and BASFI (*p* < 0.001)
**Xie**	China	2019	23	23	18–60 *	Baduanjin Qigong exercise	First phase: twice per week for 4 weeks—Second phase: 3 times per week for 8 weeks	Alternative	NA	32	Usual care	yes	yes	yes	no	no	No difference between groups

Legend: IG = Intervention Group, CG = Control Group, ROM = Range of Motion, US = Ultrasound, * mean age or age range if mean age not available, ** Results directly obtained from the authors.

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
