# Peer review of "Effectiveness of Physiotherapy in Patients with Ankylosing Spondylitis: A Systematic Review and Meta-Analysis"

_healthcare, 2022, doi:10.3390/healthcare10010132_

Round 1

Reviewer 1 Report

The article entitled Effectiveness of physiotherapy in patients with ankylosing spondylitis: a systematic review and meta-analysis aims to study whether supervised physiotherapy is effective in reducing symptomatology and improving quality of life in patients with AS compared to usual care or home-based exercise programs. It is a well-developed, methodologically high-quality and well-written article. 

Introduction: 
The authors provide an extensive introduction, with a good literature review and justification of the work. Suggestions for improvement: 

  1. There are really old bibliographic references. In a systematic review with meta-analysis, I understand that one should go for the most current evidence possible. Of course, I understand that there are citations whose reference should be what it is, but I would advise updating those that can be updated. For example, there are articles on epidemiology that are much more up to date than those provided.
  2. I would recommend trying to unify many of the sentences as if they were paragraphs, in paragraphs of more than one sentence, to help the reader. 

Material and methods
The authors have carried out a rigorous methodology. Only one comment: 

  1. The date of the last search was 27 November 2020. Given that more than a year has passed since then, have you checked if any new papers have appeared during this time?
  2. Include in the analysis of the results that a publication bias analysis has been performed using the funnel plot. 

Results: 

  1. Add in Figure 1, in the articles excluded in the qualitative synthesis, why the articles are excluded and the number of articles excluded in the quantitative synthesis. In the text it says "Four were excluded because of supervised physiotherapy in intervention and 177 control groups", but it is not clear why this exclusion was made.
  2. Correct figure 2 to homogenise the size of all the studies.
  3. The sentence: "For all other comparisons, publication bias was not assessable according to 13.3.5.4 209 Tests for funnel plot asymmetry.", line 209, is not understood.
  4. In the funnel plot there seems to be some evidence of asymmetry in both, why do the authors claim that there is no asymmetry in BASFI? 

Discussion: Perhaps the discussion is the one that needs to be improved in depth. There are many ideas, very different from each other, without being united or cohesive. Also, there is a lot of repetition of results. 

  1. 1. I would recommend, as in the introduction, not to make one-sentence paragraphs. 
    2.    I think that the paragraph from lines 275-284 does not make sense to repeat it again, as it seems to be very similar to the results already exposed. 
    3.    Similarly, the paragraphs on lines 285 and 288 say the same thing. 
    4.    Similarly, the paragraphs on lines 297, 302, 305 repeat the results. 
    Justify the sentence: "Despite all patients being aware about the necessity of daily physical exercise, most of them tend not to participate in exercise on a frequent basis", lines 321-323. On which reference do the authors rely?
    6.    I think that an interpretation of the results should be made. 
    7.    I think a paragraph or section should be added in the discussion of clinical implications. What is the best treatment for these patients? For how long? If a clinician, a physiotherapist picks up the paper, they should be able to read in a paragraph what is best. 
    8.    In the limitations section, the publication bias detected and the high heterogeneity of all the meta-analyses performed should be discussed and justified. 
    9.    In the discussion it is mentioned that the GRADE tool has been used, but it has not been mentioned until now. Why? And if it has been used, why not provide all the information?
    10.    The conclusions repeat limitations and results, rather than being conclusions as such. 

I hope these comments will help to strengthen the article. 

Author Response

Dear Reviewer,

Thank you for your positive feedback and suggestions. We tried to follow precisely your recommendations to improve the quality of our work.

Following, we reported a point-by-point response. We hope that the article is now suitable for publication.

Introduction: 
The authors provide an extensive introduction, with a good literature review and justification of the work. Suggestions for improvement: 

  1. There are really old bibliographic references. In a systematic review with meta-analysis, I understand that one should go for the most current evidence possible. Of course, I understand that there are citations whose reference should be what it is, but I would advise updating those that can be updated. For example, there are articles on epidemiology that are much more up to date than those provided.

Authors’ response: Thank you for this suggestion. We have updated references and provided more precise information as those related to the male:female ratio (3:1 in the current literature and no more 5:1 as previously reported).

  1. I would recommend trying to unify many of the sentences as if they were paragraphs, in paragraphs of more than one sentence, to help the reader. 

Authors’ response: Thank you for your suggestion. We have unified many sentences in paragraphs separating paragraphs for specific topics to improve readability.

Material and methods
The authors have carried out a rigorous methodology. Only one comment: 

  1. The date of the last search was 27 November 2020. Given that more than a year has passed since then, have you checked if any new papers have appeared during this time?
  1. Authors’ response: We implemented our search strategy again. Four articles seemed suitable to be included - Kabul et al. (2021), Singh et al. (2021), Chen et al. (2021), and Nolte et al. (2021) – but none of them was included in our analysis because they did not meet inclusion criteria.

  1. Include in the analysis of the results that a publication bias analysis has been performed using the funnel plot. 

Authors’ response: Thank you for your suggestion, but we modified the meta-analysis following another reviewer’s comment. Mainly, we had to exclude four articles (Gurcay et al. 2008, Turan et al. 2014, Hsieh et al. 2014, and Widberg et al., 2009).

Results: 

  1. Add in Figure 1, in the articles excluded in the qualitative synthesis, why the articles are excluded and the number of articles excluded in the quantitative synthesis. In the text it says "Four were excluded because of supervised physiotherapy in intervention and 177 control groups", but it is not clear why this exclusion was made.

Authors’ response: In the PRISMA flowchart, we summarized reasons for exclusion. We excluded six studies because supervised physiotherapy was performed in the intervention and control groups. Another was excluded because both groups received different home-based exercise programs without supervised physiotherapy. The eighth study was excluded because intervention and control groups differed for Stanger bath, but both based exercise programs were given to the intervention and control groups.

  1. Correct figure 2 to homogenise the size of all the studies.

Authors’ response: We modified. Thank you.

  1. The sentence: "For all other comparisons, publication bias was not assessable according to 13.3.5.4 209 Tests for funnel plot asymmetry.", line 209, is not understood.

Authors’ response: Thank you. We have modified: “According to “13.3.5.4 tests for funnel plot asymmetry” [38] that recommends using tests for funnel plot only when there are at least ten studies included in the meta-analysis not to reduce the power of the test, publication bias was not assessed.”

  1. In the funnel plot there seems to be some evidence of asymmetry in both, why do the authors claim that there is no asymmetry in BASFI? 

Authors’ response: As previously reported, we had modified our approach to publication bias. The funnel plot was deleted.

Discussion: Perhaps the discussion is the one that needs to be improved in depth. There are many ideas, very different from each other, without being united or cohesive. Also, there is a lot of repetition of results. 

  1. I would recommend, as in the introduction, not to make one-sentence paragraphs. 

Authors’ response: We have modified.

  1.    I think that the paragraph from lines 275-284 does not make sense to repeat it again, as it seems to be very similar to the results already exposed. 

Authors’ response: We have modified.

  1.    Similarly, the paragraphs on lines 285 and 288 say the same thing. 

Authors’ response: We have modified.

  1.    Similarly, the paragraphs on lines 297, 302, 305 repeat the results. 

Authors’ response: We have modified.

  1. Justify the sentence: "Despite all patients being aware about the necessity of daily physical exercise, most of them tend not to participate in exercise on a frequent basis", lines 321-323. On which reference do the authors rely?

Authors’ response: We have added proper citation (Passalent et al. 2010)

  1.    I think that an interpretation of the results should be made. 

Authors’ response: We have added a paragraph to interpret our results.

  1.    I think a paragraph or section should be added in the discussion of clinical implications. What is the best treatment for these patients? For how long? If a clinician, a physiotherapist picks up the paper, they should be able to read in a paragraph what is best. 

Authors’ response: We have added this paragraph in conclusion: “From the evidence of this work, it emerged that supervised physiotherapy is an essential part of AS management to reduce symptomatology. Physiotherapists should alternate supervised physiotherapy with home-based exercise programs to implement a virtuous cycle of instruction, demonstration, and reflection.”

  1.    In the limitations section, the publication bias detected and the high heterogeneity of all the meta-analyses performed should be discussed and justified. 

Authors’ response: Thank you for this suggestion. We have added more information about limitations due to lack of publication bias test (we had to exclude some studies following another reviewer’s suggestion, so it was not possible to evaluate publication bias) and reported how we tried to handle heterogeneity due to different kinds of interventions and number of sessions.

  1.    In the discussion it is mentioned that the GRADE tool has been used, but it has not been mentioned until now. Why? And if it has been used, why not provide all the information?

Authors’ response: Sorry for these incongruences. We forgot to attach summary of findings’ tables as supplementary materials and to add GRADE in methodology and results. Now, we have added this information.

  1.    The conclusions repeat limitations and results, rather than being conclusions as such. 

Authors’ response: We have modified the conclusion highlighting clinical implications for physiotherapists.

Reviewer 2 Report

The aim of this metaanalysis was to compare the effect of supervised physiotherapy versus usual care or home based exercises  on different disease outcome measures in ankylosing spondylitis.

Sixteen studies were included, however, several among these are not suitable to be included in the metaanalysis, and hence, the obtained results are unreliable. Please consider the following:

The study of Gurcay et al 2008 was classified as intervention vs usual care, however, home based exercise programs were given to the intervention and control groups.

In the study of Turan et al. 2014 both intervention and control groups were offered physiotherapy programmes.

In the study of Hsieh et al. 2014, the investigated patient groups received different home-based exercise programmes without supervised physiotherapy.

Also, in the study of Widberg et al. 2009, the intervention group received supervised physiotherapy + a home- based exercise programme.

Author Response

Dear reviewer,

Thank you for your suggestions. As you recommended, we have excluded indicated studies. We hope to have improved the quality of the manuscript. Thank you for your time and your precious work.

Reviewer 3 Report

Thank you for the possibility to review the article entitled “Effectiveness of physiotherapy in patients with ankylosing  spondylitis: a systematic review and meta-analysis”. The article was generally well written, the argumentation is sound, and the paper well structured.

Comments

  1. The opening paragraph is too vague and does really lay a good foundation for the context of the study.
  2. The goal of the study needs to be properly highlighted and justified. Instead of setting their aim in the frame of a simple question, I would recommend that the authors attempt to present the key objectives of their study with regards to what is presently known (i.e. literature), thus highlighting the added value of the article.
  3. I recommend the authors to add the hypothesis of the study. These should be backed with theoretical considerations.
  4. I suggest authors to integrate these categorical variables in the table 1 and so in the analysis process : continent, country
  5. Please provide the main finding of each study (in the table 1, add a column “main results”)

Author Response

Dear Reviewer,

Thank you for your positive feedback and suggestions. We tried to follow precisely your recommendations to improve the quality of our work. Following, we reported a point-by-point response. For English, a native English speaker ha checked the text again. We hope that the article is now suitable for publication.

  1. The opening paragraph is too vague and does really lay a good foundation for the context of the study.

Authors’ response: Thank you for this suggestion. We have decided not to modify the opening paragraph for two reasons: the first is due to the positive feedback by another reviewer who only suggested updating citations; the second reason is due to an initial authors’ decision. In fact, given that chosen journal is not specific on our research topic, we think that an appropriate introduction accompanied with some epidemiological information may help the reader better frame the topic and its interest.

  1. The goal of the study needs to be properly highlighted and justified. Instead of setting their aim in the frame of a simple question, I would recommend that the authors attempt to present the key objectives of their study with regards to what is presently known (i.e. literature), thus highlighting the added value of the article.

Author’s response: Thank you for this suggestion that helped us to better focus on the reasons we had induced us to perform this work. To date, some systematic reviews have been conducted to investigate the effectiveness of exercise programs in improving AS symptomatology, but none of them analyzed the impact of supervised physiotherapy. We have added this explanation before to introduce the aim of our work.

  1. I recommend the authors to add the hypothesis of the study. These should be backed with theoretical considerations.

Authors’ response: Thank you for this suggestion. We have added our hypothesis.

  1. I suggest authors to integrate these categorical variables in the table 1 and so in the analysis process: continent, country

Authors’ response: we have included the country in table 1 and reported description in the results. We decided not to report continent because this data could have appeared redundant

  1. Please provide the main finding of each study (in the table 1, add a column “main results”)

Authors’ response: thank you for your suggestion. We have added the main results in table 1.

Round 2

Reviewer 2 Report

Please revise the abstract for any inaccuracies. It is mentioned in the revised version  that 16 RCT were included. They should be 12.

Author Response

Dear reviewer,

thank you. We have just modified.